# Reliability, validity and discriminability of patient reported outcomes for non-specific low back pain in a nationwide physical therapy registry: A retrospective observational cohort study

Guus A. Meerhoff[1,2]*, Arie C. Verburg[1], Renske M. Schapendonk[1,3], Juliette Cruijsberg[1], Maria W. G. Nijhuis-van der Sanden[1], Simone A. van Dulmen[1], Philip J. Van der Wees[1]

1 Radboud University Medical Center, Radboud Institute for Health Sciences, IQ Healthcare, Nijmegen, The Netherlands, 2 Royal Dutch Society for Physical Therapy (KNGF), Amersfoort, The Netherlands, 3 Dutch Health Authority (NZA), Utrecht, The Netherlands

* guus.meerhoff@radboudumc.nl

## Abstract

### Background

A national clinical registry was established in the Netherlands containing data directly sampled from electronic health record systems of physical therapists (PTs). This registry aims to evaluate the potential of patient reported outcome measures (PROMs) to develop quality indicators (QIs) in physical therapy care.

### Purpose

To test to what extent the collected PROM data are reliable, valid and discriminatory between practices in measuring outcomes of patients with non-specific low back pain (NSLBP).

### Methods

In this retrospective cohort study 865 PT practices with 6,560 PTs voluntarily collected PROM data of patients with NSLBP, using the Quebec Back Pain Disability Scale (QBPDS), the Numeric Pain Rating Scale (NPRS) and the Patient Specific Functioning Scale (PSFS). Reliability was determined by analysing the completeness of the dataset, the comparability by using national reference data, and through checking selection bias in the included patients. Validity was tested using the known-groups contrast between patients with (sub) acute vs. chronic NSLBP. To determine discriminative ability of outcomes between PT practices, case-mix corrected hierarchical multilevel analyses were performed.

### Results

Reliability was sufficient by confirming fifteen of the sixteen hypotheses: 59% of all patients opted in for data analysis, 42% of these included patients showed repeated measurement,

**Data Availability Statement:** We are happy to be able to share the anonymized dataset with PLOS ONE. This enables us to comply with the important academic FAIR principles. We have added all necessary datasets and syntaxes as Supporting Information files. In uploading the datafiles (SPSS) we had to compress (.ZIP) the files.

**Funding:** The Quality in Motion program was funded by the association KNGF (Koninklijk Nederlands Genootschap voor Fysiotherapie). The funders had no role in study design, data collection and analysis, decision to publish, or preparation of the manuscript.

**Competing interests:** The authors have declared that no competing interests exist.

comparing with reference data and potential selection bias showed < 5% between group differences, while differences between (sub)acute and chronic NSLB-groups were significantly larger than 5% (less treatment sessions, lager differences in outcomes in (sub)acute NSLB patients). In addition, all nine adjusted hierarchical multilevel models confirm that the collected dataset on outcomes in PT care is able to discriminate between practices using PROM results of patients with NSLBP (ICC-scores range 0.11–0.21).

## Limitations

Although we have shown the reliability, validity and discriminative ability of the dataset in the quest to develop QIs, we are aware that reducing missing values in patient records and the selective participation of PTs that belong to the innovators needs attention in the next stages of implementation to avoid bias in the results.

## Conclusion

PROMs of patients with NSLBP collected in the national clinical registry of KNGF are reliable, valid and able to discriminate between primary care PT practices.

## Introduction

Within healthcare, the use of health outcomes as quality indicators (QIs) to enable transparency of the service delivery is gaining momentum [1–5]. Although seemingly promising, there is still limited experience with the development and implementation of QIs based on health outcomes [6, 7]. Presumably, data collected in clinical registries can be used in developing such QIs, ultimately assisting in achieving more transparency of service delivery [8–10].

QIs are "measurement tools, screens, or flags that are used as guides to monitor, evaluate, and improve the quality of patient care, clinical support services, and organizational functions that affect patient outcomes" [11] (p 524). QIs consist of explicitly defined and measurable items referring to the structures, processes or outcomes of care [12–14], and have the potential to be used to support quality improvement, accountability and to provide transparency of service delivery in healthcare [15]. A prerequisite of QIs to be of added value, is that they are defined using data, for example from KNGF's clinical registry, that are valid, reliable and able to discriminate between groups of interest [2].

The Royal Dutch Society for Physical Therapy (KNGF) aims to develop and implement QIs using data from a clinical registry. To do so they initiated a quality program called Quality in Motion [16]. In this program a national clinical registry has been developed that enables the anonymous collection of data on: patient characteristics, structure, processes and (patient reported) health outcomes. Data are only collected after a one-off informed consent was provided by the patient and their therapist and recorded in the patient file. The data is collected directly from the electronic health record (EHR) systems used in primary care physical therapy (PT) practices [16].

Clinical practice guidelines (CPGs) often include recommendations on healthcare delivery and outcomes that can be transformed into QIs [12, 17–20]. An example of such a recommendation is the use of (patient reported) outcomes that measure health-related aspects such as physical functioning or perceived pain and can be used to evaluate treatment [21–26]. Since the KNGF has a long history in developing high quality CPGs [22, 27], they decided to use the

recommendations in their CPGs for the selection of patient reported outcome measures (PROMs) as a basis for developing QIs. PROMs were chosen for the development of QIs, since they enable the measurement of health outcomes based on the direct perception of patients and these instruments are recommended in the high quality CPGs of KNGF.

The purpose of this study was to test to what extent health outcomes collected with PROMs in the clinical registry of KNGF are reliable, valid and able to discriminate between PT practices. These psychometric properties have been tested on data collected in patients with non-specific low back pain (NSLBP), a patient category with a large prevalence in physical therapy care. The following research questions were formulated:

1. What is the reliability and validity of health outcomes in the clinical registry measured with PROMs?

2. To what extent do health outcomes of physical therapy care collected in the clinical registry discriminate between practices using PROM results of patients with NSLBP?

## Materials and methods

### Study design and setting

This is a retrospective observational cohort study based on data collected in KNGF's clinical registry by PT practices in Dutch primary care physical therapy. All data collected between 01-01-2013 until 28-11-2018 were used.

### Participants

A total of 865 PT practices with 6,560 physical therapists (PTs) voluntarily collected data of patients with NSLBP.

### Data collection

Data on all patient categories that visited the PT were collected in the registry. This was done by uploading anonymous data from patient records in the clinically used EHR-systems to KNGF's clinical registry. To ensure uniformity of the data collection [28], all data in the registry are collected based on predefined technical specifications [16]. This procedure of data collection has shown to be feasible [16].

Data were only uploaded to the clinical registry from the EHR-systems if: 1) the patient provided their informed consent for anonymous use of the clinical data from their patient record, and 2) the therapist provided their informed consent on the use of all patient records for which they received an informed consent of the patient. For the initial reliability analysis that focussed on the completeness and comparability, all data from the clinical registry were used. For the other analyses, that specifically aimed on patients with NSLBP a selection was made and data of patients were only included if: patients visited their PT due to NSLBP, the treatment episode was finished, patients were 18 years or older, if gender and the level of chronicity of their complaints was recorded (i.e. (sub)acute vs. chronic, with (sub)acute being 0–12 weeks since onset, and chronic >12 weeks [29]), and if a PT practice had collected data from at least 30 patients with NSLBP.

### Selected outcome measures

We used the PROMs that were recommended in KNGF's clinical practice guideline (CPG) for low back pain [25]. These PROMs are: the Quebec Back Pain Disability Scale (QBPDS), the Numeric Pain Rating Scale (NPRS) and the Patient Specific Functioning Scale (PSFS).

The QBPDS is a 20-item PROM which measures the domain physical functioning using a 6-point Likert scale, ranging from 0 "not difficult at all" to 5 "unable to do". The overall score on the QBPDS is the sum score of the 20 items. A minimum score (0) represents "not being disabled at all" and the maximum score (100) represents "being maximally disabled" [30, 31]. The QBPDS is a feasible PROM which takes patients approximately 10 minutes to administer. The QBPDS scores limited to moderate evidence for good reliability, validity and usability [32]. In addition, it has been identified as a responsive PROM with a minimal important change (MIC) of 20 points [32, 33].

The NPRS is an instrument that assesses the domain perceived pain intensity [34]. It is a 1-item questionnaire with an 11-point scale ranging from 0 "no pain" to 10 "extreme pain" [35, 36]. The NPRS is identified to be feasible and easy to administer [35, 36]. The NPRS scores moderate to high on the psychometric properties reliability and validity [34, 37] and is responsive with a MIC of 2 points [37, 38].

The PSFS, which is equivalent to the Patient Specific Complaints (PSC) instrument [39], measures the domain physical functioning and "involves four steps, in which the patient's main activity problems are identified, prioritized, scored and evaluated" [40] (p. 2). Each activity is scored on a 11-point scale ranging from 0 "Able to perform activity at pre-injury level" to 10 "Unable to perform activity" [41]. The PSFS is a feasible instrument with good to excellent measurement properties (reliability, validity and sensitivity to change) [38, 41]. The MIC for the PSFS was set at 2 points [42].

## Sample size

In deciding if the selected data exceeds the sample size threshold, a general rule of thumb concerning the ability to discriminate outcomes between practices states that a minimum of 30 PT practices are required which should include a minimum of 30 patients each equalling 900 patients [28, 43]. The included PROMs in this study measure the domains physical functioning, and pain intensity. Perreault and Dionne (2005) estimated the agreement between patients and PTs on the domain physical functioning (ICC = 0.56) and pain intensity (ICC = 0.55) [44]. This enabled us to calculate a more specific sample size, using the equation of Twisk (2013), presented in S1 Appendix [45]. For this study this resulted in a minimum sample size of 1963 patients, treated in 66 PT practices.

## Data analysis

**Reliability of the data.**    The reliability of the included data was determined in four ways. First, by determining the completeness of the complete dataset of the clinical registry. This was done using the mean percentage of patients that were included (opted-in) in the database from the EHR-systems. Since data collection in the clinical registry is innovative and involves the anonymous processing of personal data on health status, we assumed that not all patients were willing or invited by their therapist to cooperate, which may result in selection bias if the number of patients that did not opt-in is too high. Given the early stage of implementation of the registry, we hypothesized it to be realistic to aim for a percentage of opted-in patients that lies above 50%. In addition, using the same dataset, completeness was determined by calculating the mean percentage of the (opted-in) patient records where a repeated pre- and post-treatment measurement with one of the selected PROMs was executed. Based on previous studies in the Netherlands and Israel, we hypothesized it to be realistic to set this percentage at a minimum of 40% [16, 46].

The second evaluation of reliability was estimating comparability of patient characteristics in the opted-in population in the complete clinical registry with national reference data. We compared the characteristics *age* and *gender* of all patient records of the clinical registry with a

dataset that is considered to be the national reference data [47]. This analysis enabled us to check if the data in the registry is comparable to a dataset which is seen as a national benchmark. The age categories chosen for this study were aligned with those used in the national reference dataset.

The third evaluation estimated the potential selection bias in the dataset of patients with NSLBP selected from the clinical registry based on inclusion rates per practice. This analysis was executed by dividing the records of patients with NSLBP in the clinical registry into two groups: practices with low vs. practices with high inclusion rates. The two groups were created using the median percentage of patients per PT practice that are included (opted-in) in the registry, based on the patient's informed-consent. The group with low inclusion rates was compared to the group with high inclusion rates on the characteristics *age*, *gender* and *number of treatment sessions*. This analysis enabled us to test again if 'the percentage of opted-in patients' created selection bias in the included population, which in turn would decrease reliability.

For the fourth reliability analysis, we further estimated potential selection bias in the dataset on patients with NSLBP through a different approach. The presence or absence of a repeated measurement with one of the selected PROMs (QBPDS, NPRS or PSFS) was used to create two groups. The group with a repeated measurement was compared to the group without a repeated measurement on the characteristics *age*, *gender* and *number of treatment sessions*. This enabled us to check if 'the presence/absence of a repeated PROM measurement' created selection bias in the included population, which in turn would decrease reliability.

Due to the large sample sizes it was expected that, using a t-test, the differences between the groups for the second, third and fourth reliability analyses would be statistically significant on all items. Nevertheless, we hypothesized that the analyses would not result in relevant differences between the created groups. We set an a priori threshold of >5% difference as relevant, i.e. groups were considered equal if the differences were ≤5%. Differences between the created categories were established by calculating the relative differences on the outcomes of the created groups.

**Validity of the data.** Regarding the validity of the collected data on patients with NSLBP the known-groups validity was determined, which is a component of construct validity frequently used to determine psychometric properties of measurement instruments [48]. With known-groups validity, the ability to distinguish or discriminate among distinct groups is defined [49]. We expected that distinct groups in the clinical registry would be present, and we defined the known-groups to be patients with chronic NSLBP versus those with (sub)acute NSLBP. Based on the findings of Costa et al. (2012) we hypothesized that patients with chronic NSLBP would need a higher number of *treatment sessions* and would achieve a lower *change score on the three PROMs of interest (QBPDS, NPRS and PSFS)* [29]. Due to the large sample sizes it was expected that, using a t-test, the differences between the groups would be statistically significant on all items, therefore we set an a priori threshold of >5% difference between the groups as being relevant. Differences between the created categories were established by calculating the relative differences on the outcomes of the created groups. For this analysis only the records of patients with NSLBP were included in which data of at least one of the selected PROMs was available at baseline ($T_0$) and endpoint ($T_{end}$) of treatment.

**The ability to discriminate between practices.** To determine the ability of the collected dataset on patients with NSLBP in the clinical registry to discriminate outcomes between PT practices using the PROM results, several hierarchical multilevel analyses were performed. Initially, for each of the outcome measures -QBPDS, NPRS and PSFS- three intercept-only hierarchical multilevel models were estimated. Each model resulted in an intraclass correlation coefficient (ICC), representing the ability of the collected data to discriminate outcomes between PT practices. If an ICC is >0.10 it can be interpreted as adequate, indicating that the model is able to discriminate outcomes between PT practices [50]. The ICC values in

discriminating outcomes of healthcare typically range between 0.05 and 0.20 [51]. The three different models were estimated since they represent the possible outcomes of the collected PROMs. The first model estimated the mean pre-posttreatment ($T_0$—$T_{end}$) change scores with 95% confidence intervals (CI) of individual practices. The second model estimated the mean percentage and 95% CI of patients in which the MIC was achieved of individual practices. The third model estimated the mean post-treatment score ($T_{end}$) with 95% CI on the selected PROMs of individual practices. For each of the three models overall mean scores for all practices combined were also estimated.

Next, it was tested if stronger models were created when adjusted models were estimated, applying case-mix correction for the independent variables: *age*, *gender*, *chronicity of the complaints* (i.e. (sub)acute vs. chronic, with (sub)acute being 0–12 weeks since onset and chronic >12 weeks) and *severity of complaints at the start of the treatment* (using the results on the PROMs). These independent variables were seen as contextual variables that may influence the results on the outcome measures. As defined by Twisk (2019) [51], all such variables can be included in the model at once [51]. The ICC presents "a good gauge of whether a contextual variable has a significant effect on the outcome" [52] (p 818). The model with the highest ICC value represents the strongest model and was selected to estimate the ability of the collected data in the clinical registry to discriminate outcomes between PT practices.

Differences between groups for reliability analyses were statistically tested using independent-sampled t-tests or Fisher's Exact test if the assumption on normality was met. IBM SPSS Statistics for Windows, version 23 (IBM Corp., Armonk, NY) was used for all analyses.

### Ethical considerations

This study was conducted according to the principles of the Declaration of Helsinki (version October 2013) and in accordance with the Medical Research Involving Human Subjects Act (WMO). The study protocol was approved by the regional Medical Ethical Committee of Radboud university medical center (registration #2014/260).

## Results

Overall the clinical registry contained 213,245 records of patients with NSLBP, collected from 865 PT practices, see S1 Fig for a flowchart of the patient inclusion. The mean age of the patients was 52.4 years (SD = 17.2), 55.0% of the patients was female and 77.5% of the patients had (sub) acute complaints. The registry contained 21,758, 54,904 and 73,554 patient episodes with repeated measurements of the QBPDS, NPRS and PSFS, respectively (See Table 1). See Table 2 for a summary on all executed analyses on the different psychometric properties which are explained below.

**Table 1. Descriptive statistics of mean PROM-scores (unadjusted for T0 scores) at practice level.**

|  | QBPDS* | NPRS# | PSFS^ |
|---|---|---|---|
|  | (score range: 0–100) | (score range: 0–10) | (score range: 0–10) |
| PT practices (N) | 204 | 405 | 500 |
| Patients (N) | 21,758 | 54,904 | 73,554 |
| Mean baseline score (SD) | 40.0 (18.4) | 6.3 (1.8) | 6.9 (1.9) |
| Mean end score (SD) | 13.1 (15.9) | 2.2 (2.1) | 1.9 (2.4) |
| Mean change T0-Tend (SD) | -27.0 (19.8) | -4.2 (2.5) | -5.0 (2.7) |

* QBPDS–Quebec Back Pain Disability Scale.

# NPRS–Numeric Pain Rating Scale.

^ PSFS—Patient Specific Functioning Scale.

**Table 2. A summary on all executed analyses on the different psychometric properties.**

| Psychometric property | Aspect of interest | Determined by: |
|---|---|---|
| Reliability | Completeness | • Calculating the percentage of patients that were opted-in. It was hypothesized to be realistic to aim for a percentage of opted-in patients that lies above 50%. |
| | | • Calculating the mean percentage of the (opted-in) patient records where a repeated pre- and post-treatment measurement with one of the selected PROMs was executed. It was hypothesized to be realistic to set this percentage at a minimum of 40%. |
| | Comparability | Comparing the patient characteristics *age* and *gender* of all patient records of the clinical registry with a dataset that is considered to be the national reference data. It was hypothesized that the analyses would not result in relevant differences. |
| | Selection bias | • Dividing the records of patients with NSLBP in the clinical registry into two groups: practices with low vs. practices with high inclusion rates. The two groups were created using the median percentage of patients per PT practice that are included (opted-in) in the registry, based on the patient's informed-consent. The group with low inclusion rates was compared to the group with high inclusion rates on the characteristics *age*, *gender* and *number of treatment sessions*. It was hypothesized that the analyses would not result in relevant differences between the created groups. |
| | | • Dividing the records of patients with NSLBP in the clinical registry into two groups based on the presence or absence of a repeated measurement with one of the selected PROMs (QBPDS, NPRS or PSFS). The groups were compared on the characteristics *age*, *gender* and *number of treatment sessions*. It was hypothesized that the analyses would not result in relevant differences between the created groups. |
| Validity | Known-groups validity | Determined by analysing the differences on the variables *number of treatment sessions* and *achieved change score on the used PROMs*. It was hypothesized that patients with chronic NSLBP would need a higher number of *treatment sessions* and would achieve a lower *change score on the three PROMs of interest (QBPDS, NPRS and PSFS)*. |
| Discriminant ability | Intraclass Correlation Coefficients | Executing several hierarchical multilevel analyses, both intercept-only and adjusted models. If an ICC is >0.10 it can be interpreted as adequate, indicating that the model is able to discriminate outcomes between PT practices. |

## Reliability of the collected data

The analysis on the completeness, using the complete dataset of the clinical registry, showed that 59.2% of the patients provided permission for the use of their data, thus are opted-in. In total 41.7% of the complete dataset of the clinical registry has executed a repeated measurement with one of the selected PROMs. Both percentages exceed the benchmark that we have defined, confirming our hypotheses on completeness.

Comparing data from the complete dataset of the clinical registry with the national reference data showed a statistically non-significant difference in gender of 0.2%, and significant differences in all age groups with percentages between -2.3% and 2.5% (see Table 3). None of the comparisons exceeded the a priori defined threshold of 5% for being relevant differences, confirming our hypothesis on comparability.

Estimating the potential selection bias in the included patients with NSLBP, the analysis comparing practices with high versus low inclusion rates of patients, resulted in non-significant differences of -0.2%, 1.1% and -2.8% on age, gender and number of treatment sessions, respectively (see Table 4). The analysis comparing presence or absence of a repeated measurement with one of the selected PROMs, using the same dataset, resulted in statistically significant differences, of 2.7%, -0.2% and 35.0% on age, gender and number of treatment sessions, respectively (see Table 5). Five of the six analyses confirmed our hypotheses on selection bias since they did not exceed the a priori defined threshold of 5%.

## Known-groups validity of the collected data

Patients with (sub)acute NSLBP needed 25.7% less treatment sessions than patients with chronic NSLBP. The three other analyses showed that patients with (sub)acute NSLBP compared to patients with chronic NSLBP achieved a 40.1%, 29.4% and 23.8% higher change score on the QBPDS, NPRS and the PSFS, respectively. The results of these four analyses for the

**Table 3. Reliability analysis on comparability.**

|  | **Total clinical registry** | **National reference data [47]** | **Difference (%)***|
|---|---|---|---|
| *Age distribution*: |  |  |  |
| PT practices (N) | 1,812 | N/A | N/A |
| Patients (N) | 1,377,215 | 29,326 | N/A |
| Patients aged 0–4 (N (%)) | 9,522 (0.7) | 59 (0.2) | 0.5[†] |
| Patients aged 5–17 (N (%)) | 87,125 (6.3) | 1,438 (4.9) | 1.4[†] |
| Patients aged 18–44 (N (%)) | 404,583 (29.4) | 7,8892 (6.9) | 2.5[†] |
| Patients aged 45–64 (N (%)) | 499,483 (36.3) | 10,3813 (5.4) | 0.9[†] |
| Patients aged 65–74 (N (%)) | 210,511 (15.3) | 4,8091 (6.4) | 1.1[†] |
| Patients aged 75–84 (N (%)) | 126,351 (9.2) | 3,372 (11.5) | -2.3[†] |
| Patients aged ≥ 85 (N (%)) | 39,640 (2.9) | 1,378 (4.7) | 1.8[†] |
| *Gender distribution*: |  |  |  |
| PT practices (N) | 1,812 | N/A | N/A |
| Patients (N) | 1,399,926 | 29,326 | N/A |
| Male patients (N (%)) | 562,632 (40.2) | 11,730 (40.0) | 0.2 |
| Female patients (N (%)) | 837,294 (59.8) | 17,596 (60.0) | -0.2 |

All data of the KNGF's clinical registry compared to the national reference data [47].

* Percentual differences are calculated by subtracting the percentages of the *National reference data* from the *Total clinical registry data*.

†p ≤ 0.001 (t-test).

N/A: Not applicable.

known-groups validity all exceeded the set threshold of 5%, confirming our hypothesis on the known-groups validity of the data (see Table 6).

## The ability to discriminate between practices based on the collected data

Seven of the nine ICC-scores obtained from the intercept-only hierarchical multilevel analyses exceed the a priori formulated threshold of 0.10 (see Table 7). In the adjusted-models all nine ICC-scores exceeded this threshold with scores ranging from 0.11–0.21. These results indicate that the adjusted-models are able to discriminate outcomes between PT practices, confirming the discriminative ability of the data [50].

## Discussion

Overall the results of this study show that the analyses regarding the reliability of the data in the clinical registry were in line with our a priori formulated hypotheses: the data was

**Table 4. Reliability analysis on selection bias using the NSLBP sample (N = 213,245 patients from 865 PT practices) of the total clinical registry: A within groups comparison on descriptive statistics using the median opt-in to divide the NSLBP sample into two groups.**

|  | **NSLBP-sample of the total clinical registry scoring below median^ opt-in** | **NSLBP-sample of total clinical registry scoring above median^ opt-in** | **Difference (%)*** |
|---|---|---|---|
| Mean age patients (SD) | 52.6 (4.6) | 52.7 (4.9) | -0.2 |
| Percentage of female patients (SD) | 55.5 (8.1) | 54.9 (7.7) | 1.1 |
| Mean number of treatment sessions (SD) | 6.9 (2.8) | 7.1 (2.6) | -2.8 |

^ median % patients included = 68.3.

* Percentual differences are calculated using the following formula: *Difference (%)* = ((*NSLBP-sample of the total clinical registry scoring below median^ opt-in / NSLBP-sample of total clinical registry scoring above median^ opt-in*) * 100) -100.

**Table 5. Reliability analysis on selection bias using the NSLBP sample of the total clinical registry: A within groups comparison using the availability of a repeated measurement with a PROM (NPRS/ QBPDS/ PSFS) to divide the NSLBP sample in two groups.**

| | With pre- and post-test PROM use | Without pre- and post-test PROM use | Difference (%)* |
|---|---|---|---|
| PT practices (N) | 775 | 865 | N/A |
| Patients (N) | 88,852 | 124,393 | N/A |
| Mean age patients (SD) | 53.2 (17.1) | 51.8 (17.2) | 2.7[†] |
| Percentage of female patients (SD) | 55.6 (5.7) | 55.7 (6.1) | -0.2[†] |
| Mean number of treatment sessions (SD) | 8.1 (8.9) | 6.0 (8.3) | 35.0[†] |

* Percentual differences are calculated using the following formula: *Difference (%) = ((With pre- and post-test PROM use / Without pre- and post-test PROM use) * 100)* -100.

[†]p ≤ 0.001 (T-test).

N/A: Not applicable.

sufficiently complete, comparable and we did not identify selection bias in the patients that were included in the registry. All analyses on the known-groups validity met the a priori formulated hypotheses. Based on these results it can be concluded that both reliability and known-groups validity are confirmed. In addition, the hierarchical multilevel analyses confirm that the collected dataset on outcomes in PT care is able to discriminate between practices using PROM results of patients with NSLBP.

Only one of the six analyses aiming to check for selection bias in the included patients, as part of the reliability analyses, did not meet the a priori formulated hypothesis. This analysis represented the number of treatment sessions required in patients with a repeated PROM measurement versus those without. The results showed that patients who required fewer treatments more often did not complete a repeated PROM measurement. This might be explained by the fact that PT's and/or patient's belief that the completions of a repeated PROM measurement have no added value for this patient category with a short treatment period and fast recovery.

To the best of our knowledge, within the field of physical therapy similar studies, which execute a so-called practice test, aiming to investigate the reliability, validity and discriminative ability of the collected data in a clinical registry, have not been published. In several countries and healthcare settings similar initiatives started the collection of PROMs scores in clinical

**Table 6. Validity analysis on the NSLBP sample of the total clinical registry, using the known-groups validity to divide the NSLBP-sample in two groups.**

| | Patients with (sub)acute NSLBP | | | Patients with chronic NSLBP | | | Difference (%) [¤] |
|---|---|---|---|---|---|---|---|
| | Mean (SD) | PT practices (N) | Patients (N) | Mean (SD) | PT practices (N) | Patients (N) | |
| Number of treatment sessions | 7.5 (7.7) | 526 | 65,284 | 10.1 (11.3) | 472 | 18,658 | -25.7[†] |
| Change score PROM T$_0$ versus T$_{end}$: | | | | | | | |
| • QBPDS* | -28.6 (19.7) | 204 | 17,399 | -20.3 (19.0) | 186 | 4,359 | 40,1[†] |
| • NPRS[#] | -4.4 (2.5) | 405 | 43,467 | -3.4 (2.6) | 350 | 11,447 | 29.4[†] |
| • PSFS[^] | -5.2 (2.7) | 500 | 57,284 | -4.2 (2.9) | 439 | 16,270 | 23.8[†] |

[¤] Percentual differences are calculated using the following formula: Difference (%) = ((Patients with (sub)acute NSLBP Mean (SD) / Patients with chronic NSLBP Mean (SD)) * 100) -100

[†]p ≤ 0.001 (t-test).

* QBPDS–Quebec Back Pain Disability Scale.

[#] NPRS–Numeric Pain Rating Scale.

[^] PSFS—Patient Specific Functioning Scale.

**Table 7. The intraclass correlation coefficients for the intercept-only model and adjusted model on the mean change-score, the mean percentage of MIC achieved-score and the mean end score of all selected PROMs.**

| | ICC[†] intercept-only model | ICC[†] adjusted-model |
|---|---|---|
| Mean change score: | | |
| NPRS[#] | 0.10 | 0.16 |
| QBPDS[*] | 0.12 | 0.21 |
| PSFS[^] | 0.11 | 0.17 |
| Mean percentage MIC achieved score: | | |
| NPRS[#] | 0.11 | 0.14 |
| QBPDS[*] | 0.12 | 0.19 |
| PSFS[^] | 0.12 | 0.14 |
| Mean end score: | | |
| NPRS[#] | 0.12 | 0.13 |
| QBPDS[*] | 0.10 | 0.11 |
| PSFS[^] | 0.12 | 0.14 |

[†] ICC–Intraclass Correlation Coefficients.

# NPRS–Numeric Pain Rating Scale.

[*] QBPDS–Quebec Back Pain Disability Scale.

^ PSFS—Patient Specific Functioning Scale.

registries, as a prerequisite for the development of QIs based on clinical data. A first example of such registry is initiated by the National Health Service in the United Kingdom, which introduced their national PROMs programme in 2009. A second example is initiated in 2010 by the Dutch Institute for Clinical Auditing (DICA). DICA manages 22 clinical registries, belonging to different scientific associations of medical specialists (e.g. neurologists & oncologists) [53]. Despite this experience in building clinical registries, several publications suggest that the integrated implementation of PROMs, using them to stimulate shared decision-making on clinical level and as performance information on managerial level has not yet been successfully achieved [8, 54].

In the quest for developing QIs using data from clinical registries, the execution of a practice test to evaluate the psychometric properties of the collected data is crucial. There are a limited number of studies in other fields than PT that have conducted such a practice test. Examples of such studies are analysed in the review on the validity of QI's on the readmission rate of Fischer et al. (2012). This study showed that only a small proportion 21 of the 486 included papers test the actual validity of the data in some sort of practice test [55]. In addition, a systematic review of Langendam et al. (2020) showed that performing a practice test to validate the formulated QI which are based on recommendations of CPGs is relatively rare [20].

There are some publications providing a framework on the development of QIs [18, 56]. These frameworks also emphasize the importance of the execution of a practice test, since it enables formulating a benchmark for a QI that meets all psychometric criteria [2, 28, 56]. Unfortunately, several studies state that there is no standard definition of what a practice test has to contain [18, 20]. Often only the Delphi methodology is used to reach consensus in formulating QIs and practice tests are not executed [17, 19, 42, 57].

In short, there is no guidance available for applying a practice test as we have done in this study. The results of this study are therefore a good starting point for the development of standards for a practice test to be carried out in the development of benchmarks for QIs based on data from a clinical registry that meet all relevant psychometric properties.

## Strengths and limitations

The strength of this paper is the large number of PTs that voluntarily participated and collected data on a very large number of participating patients.

Nevertheless, this paper is subject to several limitations. Despite the fact that the infrastructure of the clinical registry has been successfully built up, we identified that input fields from the patient files in the EHR-systems are still registered as missing values, and thus could not be included in this study. These missing values sometimes occurred due to technical omissions (e.g. there are several QBPDS questionnaires defined in the EHR-software but only one of the questionnaires is eligible for extraction to the registry), but also due the submission of incompletely registered patient files from the EHR-system to the national registry. In the further implementation of the clinical registry, we make a continuous effort in trying to decrease the number of missing values. A specific challenge is the large number of different EHR-suppliers that is involved. We have improved the collaboration with the different EHR-system suppliers, which assists in overcoming the technical omissions. In addition, we have executed different projects trying to influence the behaviour of PTs in completing all fields of a patient file before uploading. Examples of such projects are: the organization of several regional symposia to teach PTs on how to use the registry and the development of an online (free for use) feedback dashboard. Such implementation activities are important in the further exploitation of a clinical registry. After all, having a registry with complete patient files is crucial in the development of valid benchmarks for QIs. A second limitation is the chance of selection bias based on the PT practices that participate in the data collection. This is likely to have happened since -in this early stage of implementation- the current users of the clinical registry, the PTs who voluntarily provided all data, are mainly the innovators, early adopters and early majority [58]. These participants are probably PTs who favour the use of PROMs. A third limitation might be the fact that we have used somewhat older data from the registry. Despite this, we assume that the data are still relevant as a good representation of the clinical practice and that they shed light on the potential that a clinical registry has in formulating QIs and their benchmarks. Moreover, the outcomes can be used as baseline measurement to evaluate changes over time.

## Implications for future research and clinical practice

Future research should focus on several topics. First, the further development of a standardised practice test, which is an important step before defining QIs and their benchmarks. Such a test should finally include all psychometric properties that need to be tested in defining QIs. These aspects are: reliability, validity, discriminative ability, responsiveness to change, feasibility and usability [2, 28]. The development of such a practice test will likely help with the implementation of QIs in clinical practice and will minimize the resistance in the use of QI and their benchmark. Second, research should focus on the development of actual benchmark based on the collected data thus far. This means that an extensive developmental process must be completed before a QI, including a psychometrically sound benchmark is finished. Given the investment needed it does not seem feasible to develop QIs for all conditions in the field of PT. Therefore it is relevant to investigate what solution can be found for this problem. A possible solution might be the development of several generic QIs for the field of PT and only develop specific QIs for conditions that are seen on a very regular basis by all PTs, as is the case in patients with NSLBP.

In addition, from a clinical practice perspective, efforts should be made to assist PTs in completing their patient files (e.g. by using modern technology such as applications for mobile devices which enable patients to fill in PROMs on their mobile phones) and to increase the number of PTs who will provide their data to the clinical registry. Finally, continuous

investments must be made in improving the infrastructure of the clinical registry, on the one hand to detect errors (e.g. missing values) and on the other to improve efficiency and ease of use.

## Conclusions

This study showed that the health outcomes of patients with NSLBP collected with PROMs in the national clinical registry of KNGF are reliable, valid and are able to discriminate between primary care PT practices.

## Supporting information

**S1 Appendix. The calculation used to define a more specific sample size, using the equation of Twisk et al. (2013).**
(DOCX)

**S1 Fig. A flowchart of the patient inclusion.**
(DOCX)

**S1 Syntax. 20180309_cleaning_definitief_GUUS_JC.sps.**
(SPS)

**S2 Syntax. 20201603 descriptives and provisionary analysis and QI analysis.sps.**
(SPS)

**S1 Dataset. 20180116_Totaalbestand_aanlever.sav.**
(ZIP)

**S2 Dataset. werkbestand_20180308_DCSPH.sav.**
(ZIP)

**S3 Dataset. werkbestand_20180308_DCSPH_leeftijd.sav.**
(ZIP)

**S4 Dataset. werkbestand_20180308_DCSPH_leeftijd_geslacht.sav.**
(ZIP)

**S5 Dataset. werkbestand_20180308_DCSPH_leeftijd_geslacht_afgesloten.sav.**
(ZIP)

**S6 Dataset. werkbestand_20180308_DCSPH_leeftijd_geslacht_afgesloten_duurklachten. sav.**
(ZIP)

**S7 Dataset. 20180314_multilevel.sav.**
(ZIP)

**S8 Dataset. One sample T-test Nivel data vs LDF data.sav.**
(ZIP)

## Acknowledgments

We would like to acknowledge all patients, physical therapists and practices who voluntarily collected the valuable data into KNGF's clinical registry over the past years. In addition, we would like to thank our research colleagues Marjo Maas, Janine Liefers and Annick Bakker-Jacobs for all the time they have invested over the past years in training PTs on how to work

with the registry and in their role in analysing the data which results in valuable information for the academic, the clinicians and ultimately for the patient.

## Author Contributions

**Conceptualization:** Guus A. Meerhoff, Arie C. Verburg, Renske M. Schapendonk, Simone A. van Dulmen, Philip J. Van der Wees.

**Data curation:** Guus A. Meerhoff, Arie C. Verburg, Renske M. Schapendonk, Juliette Cruijsberg, Maria W. G. Nijhuis-van der Sanden, Philip J. Van der Wees.

**Formal analysis:** Guus A. Meerhoff, Arie C. Verburg, Renske M. Schapendonk, Juliette Cruijsberg.

**Funding acquisition:** Guus A. Meerhoff, Maria W. G. Nijhuis-van der Sanden, Simone A. van Dulmen, Philip J. Van der Wees.

**Investigation:** Guus A. Meerhoff, Renske M. Schapendonk, Maria W. G. Nijhuis-van der Sanden, Simone A. van Dulmen, Philip J. Van der Wees.

**Methodology:** Guus A. Meerhoff, Arie C. Verburg, Maria W. G. Nijhuis-van der Sanden, Simone A. van Dulmen, Philip J. Van der Wees.

**Project administration:** Guus A. Meerhoff, Renske M. Schapendonk, Simone A. van Dulmen, Philip J. Van der Wees.

**Resources:** Guus A. Meerhoff, Simone A. van Dulmen, Philip J. Van der Wees.

**Software:** Guus A. Meerhoff, Renske M. Schapendonk.

**Supervision:** Guus A. Meerhoff, Maria W. G. Nijhuis-van der Sanden, Simone A. van Dulmen, Philip J. Van der Wees.

**Validation:** Guus A. Meerhoff, Arie C. Verburg, Simone A. van Dulmen.

**Visualization:** Guus A. Meerhoff.

**Writing – original draft:** Guus A. Meerhoff.

**Writing – review & editing:** Guus A. Meerhoff, Arie C. Verburg, Renske M. Schapendonk, Maria W. G. Nijhuis-van der Sanden, Simone A. van Dulmen, Philip J. Van der Wees.

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
