## [Decision Letter · Decision Letter 0]

19 Mar 2021

PONE-D-21-03296

Reliability, validity and discriminability of patient reported outcomes for non-specific low back pain in a nationwide physical therapy registry: a retrospective observational cohort study

PLOS ONE

Dear Dr. Meerhoff,

Thank you for submitting your manuscript to PLOS ONE. After careful consideration, we feel that it has merit but does not fully meet PLOS ONE’s publication criteria as it currently stands. Therefore, we invite you to submit a revised version of the manuscript that addresses the points raised during the review process.

Please address the points raised by the two reviewers.

We look forward to receiving your revised manuscript.

Kind regards,

Alison Rushton

Academic Editor

PLOS ONE

Journal Requirements:

Reviewers' comments:

Reviewer's Responses to Questions

**Comments to the Author**

1. Is the manuscript technically sound, and do the data support the conclusions?

Reviewer #1: Yes

Reviewer #2: Yes

2. Has the statistical analysis been performed appropriately and rigorously? 

Reviewer #1: Yes

Reviewer #2: Yes

3. Have the authors made all data underlying the findings in their manuscript fully available?

Reviewer #1: No

Reviewer #2: Yes

4. Is the manuscript presented in an intelligible fashion and written in standard English?

Reviewer #1: Yes

Reviewer #2: Yes

5. Review Comments to the Author

Reviewer #1: 

Congratulations to the authors for such a good work. I would like to share some comments/seek for some clarification.

1. Suggest authors to expand on the importance and the use of quality indicators (QIs) in health services. It is unclear in the introduction why authors want to identify if the PROMs can discriminate PT practices.

2. Table III - Suggest to have the total N in the top row and present the data in the format of N(%). Suggest to use age categories of similar gap, if not, please justify in methods section.

3. Table IV, V, VI - how does the difference being derived? For example, mean age difference has a difference in %. Suggest author to include the explanation in "Methods" section.

4. Kindly specify test used to obtain p-values for Table III - VI. Is it t-test or Fisher's exact test?

5. Please provide the number of PT practices that were included in the analysis. Authors only described the number of patients .

Reviewer #2: 

Comments to the authors

Manuscript title:

Reliability, validity and discriminability of patient reported outcomes for non-specific low back pain in a nationwide physical therapy registry: a retrospective observational cohort study

Thank you for inviting me to review the manuscript which aimed at testing to what extent health outcomes in patients with non-specific low back pain (NSLBP) collected with PROMs in the national clinical registry of the Royal Dutch Society for Physical Therapy (KNGF) are reliable, valid and able to discriminate between physical therapy practices.

The paper provides background information about successful involvement of patients and physical therapists in a clinical registry in The Netherlands. The authors included data from an impressive number of physical therapy (PT) practices (n=865) and PTs (n=6,560) of patients with NSLBP over a long period of time, between 2013 and 2018. The comprehensive description of included outcome measures in addition to the methods used to answer the research questions were helpful. Findings were described in detail and discussed sufficiently.

This manuscript addresses an important issue for clinicians, researchers and other stakeholders as it demonstrates the importance and value of clinical registries also for formulating appropriate QI's and provides a good example of a practice test to assess the psychometric properties of the collected data. Therefore, it should be of interest to a wide audience.

I have some minor suggestions that could improve the manuscript further.

Abstract: The abstract summarizes the study very well. Please just correct “in this next stages of implementation”.

Introduction: The introduction is comprehensive and informs the reader about the importance of clinical registries also for the development of quality indicators. It would be interested how people consented to participate in the registry and if they have to consent only once or multiple times.

Method: described comprehensively. In this manuscript, acute vs. chronic NSLBP is defined with acute being 0-12 weeks since onset, and chronic >12 weeks. However, definitions of acute and chronic LBP are different, for example in a review provided by Chou et al. where acute NSLBP is defined <4 weeks' duration) while chronic/subacute NSLBP >4 weeks' duration. Please provide a reason for your definition and a reference.

Chou, R., & Huffman, L. H. (2007). Nonpharmacologic therapies for acute and chronic low back pain: a review of the evidence for an American Pain Society/American College of Physicians clinical practice guideline. Annals of internal medicine, 147(7), 492-504.

Results: are very well described. The tables provided help to overlook the results, but please look at the formatting of the tables, e.g. upper and lower lines in Table 1 missing. Table III By “total clinical registry” you mean all data of the KNGF’s clinical registry?

Discussion: very well written. I liked the idea of “implementation activities” to facilitate the use of the registry.

Line 295 “that the completions of a repeated PROM measurement has no added value for this patient category” should be have…

Further suggestions: Look for double spaces and commas throughout the text (e.g. after “In addition”, “Nevertheless”).

I wish the authors the best with their continued work in this interesting and promising area.

6. PLOS authors have the option to publish the peer review history of their article (what does this mean?). If published, this will include your full peer review and any attached files.

Reviewer #1: No

Reviewer #2: No

---

## [Author Response · Author response to Decision Letter 0]

28 Apr 2021

PLOS ONE

Subject: Submission of revised manuscript 

Utrecht, April 28th 2021

Dear Dr. Emily Chenette,

Thank you for the invitation to revise and resubmit our paper for publication in PLOS ONE. Please find enclosed our revised manuscript entitled: ‘Reliability, validity and discriminability of patient reported outcomes for non-specific low back pain in a nationwide physical therapy registry: a retrospective observational cohort study’.

The feedback of the reviewers was very useful. We have replied to the feedback in a point-by-point response letter including the journal requirements and the feedback from the reviewers, followed by the responses of the authors (see the Appendix). In the manuscript itself, all text revisions are visible using the track changes function of Microsoft Word (using the ‘All Markup’ mode, to keep the line numbers in correspondence with my rebuttal letter). 

We hope this revision made our manuscript eligible for publication in PLOS ONE and meets your expectations. 

We look forward to your kind consideration of our revised manuscript.

Yours sincerely,

Guus A. Meerhoff, MSc. (corresponding author)

Radboud university medical center, 114 IQ healthcare,

PO Box, 9101

6500 HB Nijmegen, The Netherlands

+ 31 6 48421927 (phone)

 

Appendix: Point by point reply to the stated journal requirements and the feedback of the reviewers

Journal requirements

Journal Requirements:

Author’s reply:

We have changed the style of our paper to align it with PLOS ONE’s style requirements.

Author’s reply:

We have checked the reference list and added references number 11-15 based on a text passage that was added in the Introduction. See also comment 1 of reviewer 1. [Line 500-511]

11. Mainz J. Defining and classifying clinical indicators for quality improvement. Int J Qual Heal Care. 2003;15(6):523-530. doi:10.1093/intqhc/mzg081

12. Westby MD, Klemm A, Li LC, Jones CA. Emerging Role of Quality Indicators in Physical Therapist Practice and Health Service Delivery. Phys Ther. 2016;96(1):90-100. doi:10.2522/ptj.20150106

13. McGlynn EA, Asch SM. Developing a clinical performance measure. Am J Prev Med. 1998;14(3 SUPPL.):14-21. doi:10.1016/S0749-3797(97)00032-9

14. Centers for Medicare and Medicaid Services. https://www.cms.gov/Medicare/Quality-Initiatives-Patient-Assessment-Instruments/QualityMeasures/index.html. Published 2019. Accessed January 16th 2021

15. Mainz JAN. Developing evidence-based clinical indicators : a state of the art methods primer. Int J Qual Heal Care. 2003;15(supplement I):5-12.

Furthermore we aligned the references with the style of PLOS ONE’s requirements (e.g. stated as a max of six authors per reference). [Line 467-642]

Author’s reply:

We are happy to be able to share the anonymized dataset with PLOS ONE. This enables us to comply with the important academic FAIR principles. We have added all necessary datasets and syntaxes as Supporting Information files. [Line 652-680]

Author’s reply:

When aligning the paper with PLOS ONE’s style requirements we have inserted the Supporting Information including its captions in the correct format.

Comments of reviewer 1

Congratulations to the authors for such a good work. I would like to share some comments/seek for some clarification.

1. Suggest authors to expand on the importance and the use of quality indicators (QIs) in health services. It is unclear in the introduction why authors want to identify if the PROMs can discriminate PT practices.

Author’s reply:

Thank you for your suggestion. We have added a paragraph providing a definition of QIs, showing their potential/importance regarding transparency of service delivery in healthcare. See the paraphrased text passage below. 

“QIs are “measurement tools, screens, or flags that are used as guides to monitor, evaluate, and improve the quality of patient care, clinical support services, and organizational functions that affect patient outcomes”. 11 (p 524) QIs consist of explicitly defined and measurable items referring to the structures, processes or outcomes of care, 12,13,14 and have the potential to be used to support quality improvement, accountability and to provide transparency of service delivery in healthcare.15” [Line 57-62]

In addition we have added a sentence in the Introduction, explaining why we want to identify if the PROMs can be used to discriminate between PT practices. PROMs are used for the development of QIs since they enable the measurement of health outcomes based on the direct perception of patients and these instruments are recommended in the Clinical Practice Guidelines (CPGs) of KNGF. 

“PROMs were chosen for the development of QIs, since they enable the measurement of health outcomes based on the direct perception of patients and these instruments are recommended in the high quality CPGs of KNGF.” [Line 79-81]

2. Table III - Suggest to have the total N in the top row and present the data in the format of N(%). Suggest to use age categories of similar gap, if not, please justify in Materials and methods section.

Author’s reply:

Thank you for your suggestion. We have changed the order of presentation in Table III. For both outcomes (‘Age distribution’ and ‘Gender distribution’) we now first present the total N (PT-practices and patients). In addition we present the data in the suggested format: N (%). [Line 293-297]

Unfortunately we are not able to change the presented age categories because we have used the age categories as used in the selected national reference data. Since we cannot change the age categorisation in this national reference data, we are obliged to use the categorisation as they are presented in their current form. We have provided the following explanation in the Materials and methods section 

“The age categories chosen for this study were aligned with those used in the national reference dataset.” [Line 180-181] 

3. Table IV, V, VI - how does the difference being derived? For example, mean age difference has a difference in %. Suggest author to include the explanation in "Materials and methods" section.

Author’s reply:

Thank you for your suggestion. In the “Materials and methods” section we have added the sentence as paraphrased below, explaining how we have derived the differences as presented in the Tables IV, V and VI. Since similar differences have been presented in Table III we have also included this explanation for this Table.

“Differences between the created categories were established by calculating the relative differences on the outcomes of the created groups.” [Line 203-205 & 218-220]

In addition we have added a footnote to the Tables, explaining how the differences were derived. [Line 295, 313-314, 319-320, 333]

4. Kindly specify test used to obtain p-values for Table III - VI. Is it t-test or Fisher's exact test?

Author’s reply:

 Thanks you for your suggestion. In the text and in the footnotes of the Tables we have specified that we used a t-test to calculate the p-values. [Line 199, 216, 296, 321]

5. Please provide the number of PT practices that were included in the analysis. Authors only described the number of patients .

Author’s reply:

Thank you for your final comment. We have provided the number of PT practices in Table I [Line 273-277], III [Line 293-297], the description of Table IV [Line 308-311] V [ Line 316-322] & VI [Line 331-337]. In addition throughout the text we have described more clearly what data selection was made for each analysis, providing insight why the numbers (of PT practices and patients) differ in the different Tables [Line: 163, 176, 180-181, 225, 265-266, 282, 284-285, 287, 293, 299]

Comments of reviewer 2

Comments to the authors

Manuscript title:

Reliability, validity and discriminability of patient reported outcomes for non-specific low back pain in a nationwide physical therapy registry: a retrospective observational cohort study

Thank you for inviting me to review the manuscript which aimed at testing to what extent health outcomes in patients with non-specific low back pain (NSLBP) collected with PROMs in the national clinical registry of the Royal Dutch Society for Physical Therapy (KNGF) are reliable, valid and able to discriminate between physical therapy practices.

The paper provides background information about successful involvement of patients and physical therapists in a clinical registry in The Netherlands. The authors included data from an impressive number of physical therapy (PT) practices (n=865) and PTs (n=6,560) of patients with NSLBP over a long period of time, between 2013 and 2018. The comprehensive description of included outcome measures in addition to the methods used to answer the research questions were helpful. Findings were described in detail and discussed sufficiently.

This manuscript addresses an important issue for clinicians, researchers and other stakeholders as it demonstrates the importance and value of clinical registries also for formulating appropriate QI's and provides a good example of a practice test to assess the psychometric properties of the collected data. Therefore, it should be of interest to a wide audience.

I have some minor suggestions that could improve the manuscript further.

1. Abstract: The abstract summarizes the study very well. Please just correct “in this next stages of implementation”.

Author’s reply:

Thank you for your correction with regard to this grammatical error. We have changed wording and used “the” instead of “this”. [Line 44]

2. Introduction: The introduction is comprehensive and informs the reader about the importance of clinical registries also for the development of quality indicators. It would be interested how people consented to participate in the registry and if they have to consent only once or multiple times.

Author’s reply:

Thank you for your suggestion to include a description on how the informed consent is obtained from both the patient and the therapist in the Introduction. We have added the following sentence in the Introduction:

“Data are only collected after a one-off informed consent was provided by the patient and their therapist and recorded in the patient file.” [Line 69-70]

Besides adding this sentence in the Introduction we felt our explanation on how the informed consent was obtained, as described in the Materials and methods section, needed some clarification. Therefore we have reworded the initial sentence in the Materials and methods section as paraphrased below:

“Data were only uploaded to the national clinical registry from the EHR-systems if: 1) the patient provided their informed consent for anonymous use of the clinical data from their patient record, and 2) the therapist provided their informed consent on the use of all patient records for which they received an informed consent of the patient.” [Line 111-114]

3. Method: described comprehensively. In this manuscript, acute vs. chronic NSLBP is defined with acute being 0-12 weeks since onset, and chronic >12 weeks. However, definitions of acute and chronic LBP are different, for example in a review provided by Chou et al. where acute NSLBP is defined <4 weeks' duration) while chronic/subacute NSLBP >4 weeks' duration. Please provide a reason for your definition and a reference.

Chou, R., & Huffman, L. H. (2007). Nonpharmacologic therapies for acute and chronic low back pain: a review of the evidence for an American Pain Society/American College of Physicians clinical practice guideline. Annals of internal medicine, 147(7), 492-504.

Author’s reply:

We understand this question. Indeed, in the literature there are different definitions for patients with (sub)acute versus those with chronic NSLBP. We choose to use the definition provided by the paper of Costa et al. since their meta-analysis from 2012 confirmed the hypothesis that there is a difference in the recovery of patients with (sub)acute versus those with chronic complaints. We have added the reference to this paper in the ‘Data collection’ paragraph in the Materials and methods section.

“…the level of chronicity of their complaints was recorded (i.e. (sub)acute vs. chronic, with (sub)acute being 0-12 weeks since onset, and chronic >12 weeks29)…” [Line 118-120]

29. Costa L da CM, Maher CG, Hancock MJ, McAuley JH, Herbert RD, Costa LOP. The prognosis of acute and persistent low-back pain: a meta-analysis. Can Med Assoc J. 2012;184(11):1229-1230. doi:10.1503/cmaj.120627

In addition we decided to be more precise in declaring the included subgroups. In accordance with the paper of Costa et al. we have changed the wording ‘acute’ to ‘(sub)acute’ throughout the entire article. 

4. Results: are very well described. The tables provided help to overlook the results, but please look at the formatting of the tables, e.g. upper and lower lines in Table 1 missing. Table III By “total clinical registry” you mean all data of the KNGF’s clinical registry?

Author’s reply:

Thank you for the provided feedback. We have checked the formatting of the Tables and changed the wording of Table III as suggested. [Line 293-297]

5. Discussion: very well written. I liked the idea of “implementation activities” to facilitate the use of the registry.

Line 295 “that the completions of a repeated PROM measurement has no added value for this patient category” should be have…

Author’s reply:

Thank you for your correction with regard to this grammatical error. We have changed wording and used “have” instead of “has”. [Line 368]

6. Further suggestions: Look for double spaces and commas throughout the text (e.g. after “In addition”, “Nevertheless”).

Author’s reply:

Thank you for your final correction. We have deleted all double spaces and added the commas when necessary.

---

## [Editor Report · Decision Letter 1]

5 May 2021

Reliability, validity and discriminability of patient reported outcomes for non-specific low back pain in a nationwide physical therapy registry: a retrospective observational cohort study

PONE-D-21-03296R1

Dear Dr. Meerhoff,

We’re pleased to inform you that your manuscript has been judged scientifically suitable for publication and will be formally accepted for publication once it meets all outstanding technical requirements.

Kind regards,

Alison Rushton

Academic Editor

PLOS ONE

Additional Editor Comments (optional):

Thank you for addressing all of the feedback from the two reviewers to a satisfactory level.
---

## [Editor Report · Acceptance letter]

25 May 2021

PONE-D-21-03296R1 

Reliability, validity and discriminability of patient reported outcomes for non-specific low back pain in a nationwide physical therapy registry: a retrospective observational cohort study 

Dear Dr. Meerhoff:

I'm pleased to inform you that your manuscript has been deemed suitable for publication in PLOS ONE. Congratulations! Your manuscript is now with our production department. 

Kind regards, 

on behalf of

Professor Alison Rushton 

Academic Editor

PLOS ONE